# Reduction-Responsive Molecularly Imprinted Poly(2-isopropenyl-2-oxazoline) for Controlled Release of Anticancer Agents

**DOI:** 10.3390/pharmaceutics12060506

**Published:** 2020-06-02

**Authors:** Michał Cegłowski, Valentin Victor Jerca, Florica Adriana Jerca, Richard Hoogenboom

**Affiliations:** 1Supramolecular Chemistry Group, Center of Macromolecular Chemistry (CMaC), Department of Organic and Macromolecular Chemistry, Ghent University, Krijgslaan 281 S4, B-9000 Ghent, Belgium; valentinvictor.jerca@ugent.be (V.V.J.); adriana.jerca@ugent.be (F.A.J.); 2 Faculty of Chemistry, Adam Mickiewicz University in Poznan, Uniwersytetu Poznańskiego 8, 61-614 Poznań, Poland; 3Centre of Organic Chemistry “Costin D. Nenitzescu”, Romanian Academy, Spl. Independentei 202B, 060023 Bucharest, Romania

**Keywords:** molecularly imprinted polymers, drug delivery, 5-fluorouracil, responsive polymer, 2-oxazoline

## Abstract

Trigger-responsive materials are capable of controlled drug release in the presence of a specific trigger. Reduction induced drug release is especially interesting as the reductive stress is higher inside cells than in the bloodstream, providing a conceptual controlled release mechanism after cellular uptake. In this work, we report the synthesis of 5-fluorouracil (5-FU) molecularly imprinted polymers (MIPs) based on poly(2-isopropenyl-2-oxazoline) (PiPOx) using 3,3′-dithiodipropionic acid (DTDPA) as a reduction-responsive functional cross-linker. The disulfide bond of DTDPA can be cleaved by the addition of tris(2-carboxyethyl)phosphine (TCEP), leading to a reduction-induced 5-FU release. Adsorption isotherms and kinetics for 5-FU indicate that the adsorption kinetics process for imprinted and non-imprinted adsorbents follows two different kinetic models, thus suggesting that different mechanisms are responsible for adsorption. The release kinetics revealed that the addition of TCEP significantly influenced the release of 5-FU from PiPOx-MIP, whereas for non-imprinted PiPOx, no statistically relevant differences were observed. This work provides a conceptual basis for reduction-induced 5-FU release from molecularly imprinted PiPOx, which in future work may be further developed into MIP nanoparticles for the controlled release of therapeutic agents.

## 1. Introduction

Cytotoxic drugs are used to treat cancer as they are primarily toxic to cells that are rapidly growing and dividing. As a result, the problems connected with their use mainly include relative lack of specificity during their biodistribution in the human body and side effects caused by the simultaneous attack of target cancer cells and healthy cells [1,2]. 5-Fluorouracil (5-FU) is a cytotoxic drug with a broad spectrum of activity against various tumors. However, due to increasing drug resistance occurring in cancer cells, its clinical use has been limited in the past 50 years. Moreover, 5-FU undergoes fast metabolism in the human body, requiring a continuous administration of high doses during therapy, which leads to the occurrence of severe toxic effects in many patients [3,4,5]. To overcome these issues, 5-FU can be used with various drug delivery systems (DDS), which release it at the desired target site. This allows for systemic toxicity to be reduced toward healthy cells and to localize 5-FU delivery only to selected tissue. Up to now, many 5-FU DDS have already been developed based on polymeric particles [6,7,8,9], hydrogels [10,11,12], magnetic nanoparticles [13,14,15], and clay minerals [16,17,18].

Molecularly imprinted polymers (MIPs) are obtained by entrapping a target template molecule within the polymer 3D network while it is being formed. Most MIPs rely on non-covalent interactions as they allow for convenient binding and release of target molecules, depending on the conditions applied [19]. These unique properties allow for the construction of MIP-based DDS imprinted with various anticancer drugs such as 5-FU [20,21], paclitaxel [22,23,24], thalidomide [25,26], sunitinib [27], and capecitabine [28].

To allow controlled release of target molecules, responsive MIPs have been introduced [29,30]. Upon application of a particular stimulus such as light, analyte, change of pH, or temperature, their structure changes, which results in increased release of the adsorbed target molecule. These properties make responsive MIPs ideal candidates for use as smart DDS, as they can release the drug in a feedback regulated way. The additional benefits of using MIPs as DDS are their high stability and durability against harsh conditions, high drug loading capacities, and easy regulation of the cross-link density or the presence of particular functionalities in their structure [19,31]. These advantages of DDS-based MIPs resulted in the construction of 2-hydroxyethylmetacrylate- and acrylic acid-based 5-FU imprinted hydrogels [32] and ethylene glycol dimethacrylate/methacrylic acid-based 5-FU MIPs [33] for prolonged release of this drug. Aside from these beneficial properties for DDS, no MIP system fulfills all requirements for DDS, indicating that there is a need to develop new polymers and monomers that can be used to obtained MIP-based DDS [34].

2-Isopropenyl-2-oxazoline (iPOx) is a versatile monomer with dual functionality that can be polymerized into poly(2-isopropenyl-2-oxazoline) (PiPOx) by radical, anionic, group transfer polymerization, and more recently controlled radical polymerization [35,36,37,38,39,40]. PiPOx is a versatile polymer possessing 2-oxazoline rings as reactive side chains that can be used for a variety of post-functionalization reactions to introduce various functional groups or to yield molecular brushes via cationic ring opening polymerization [41,42,43,44,45]. PiPOx possesses many advantages such as high hydrophilicity and biocompatibility, it is inert to moisture and oxygen, and the 2-oxazoline ring modification with (di)carboxylic acids requires simple, mild reaction conditions. PiPOx has also been shown to exhibit immune-modulative properties, thus being suited for biomedical applications [46]. Moreover, in vitro cellular studies have demonstrated the noncytotoxic character of PiPOx hydrogels cross-linked with dicarboxylic acids [47]. Recently, our group has successfully used PiPOx for the synthesis of hydrogels that were afterward used for drug delivery or water purification as well as energy dissipation applications [47,48,49]. To date, PiPOx has not been reported as a polymer precursor for the preparation of MIPs, despite its easy, robust, and straightforward cross-linking reaction with dicarboxylic acids.

Herein, we report on the synthesis of molecularly imprinted PiPOx and its application in the triggered release of 5FU. PiPOx was cross-linked in the presence of 5-FU using a dicarboxylic acid possessing a disulfide bond in its structure (3,3′-dithiodipropionic acid), which can be afterward cleaved using reductive reagents such as tris(2-carboxyethyl)phosphine) (TCEP). The introduction of disulfide bonds into poly(2-oxazoline)-based cross-linked materials has previously been reported to obtain reduction-sensitive materials [50,51]. However, there have been no previous reports on PiPOx-based cross-linked materials containing disulfides bonds. The amide groups formed during the 2-oxazoline ring opening reaction with the dicarboxylic acid were anticipated to lead to specific interactions with 5-FU. The presence of the disulfide bond in the obtained MIPs is expected to generate a reduction-triggered release in the presence of TCEP-inducing the cleavage of the MIP cavities, thus leading to increased 5-FU release. To the best of our knowledge, this is the first study describing the application of PiPOx in the synthesis of MIPs. Within this paper, we will discuss the physicochemical properties of the obtained MIPs as well as their triggered release properties.

## 2. Materials and Methods

### 2.1. Materials and Chemicals

2-Isopropenyl-2-oxazoline (Sigma-Aldrich, 98%, iPOx) was distilled over CaH_2_ under reduced pressure before use. Tetrahydrofuran (Sigma-Aldrich, THF) was freshly distilled over Na/benzophenone under Ar flow before use. *n*-Butyllithium solution 2.5 M in hexane (Sigma-Aldrich, n-BuLi) was used as received. 5-Fluorouracil (5-FU), tris(2-carboxyethyl)phosphine) (TCEP), and 3,3′-dithiodipropionic acid (DTDPA) were obtained from TCI Europe and used as such. All solvents (HPLC grade) were obtained from Sigma-Aldrich.

### 2.2. Instruments

A Bruker DMC300 (300 MHz for ^1^H, 75 MHz for ^13^C) was used to obtain nuclear magnetic resonance (NMR) spectra. UV−Vis measurements were done using a Varian Cary 100 Bio spectrophotometer. All solutions were placed in 1 cm quartz cuvettes. Calibration curves are presented in Appendix A. The absorbance of the solutions was measured at 270 nm for the DMF solutions and 266 nm for all water solutions. Samples containing higher concentrations than the ones used for calibration curve preparation were diluted prior to the measurement. A FEI Phenom desktop microscope was used to record the scanning electron microscopy (SEM) images. The polymer samples were glued using carbon tape and a gold layer was sputtered on them prior to analysis. PerkinElmer Spectrum 1000 FTIR spectrometer was used to record Fourier transform infrared (FTIR) spectra. The spectrometer was operating in the 600–4000 cm^−1^ range using attenuated total reflectance (ATR) mode. A Mettler Toledo TGA/SDTA 851e was used to investigate the thermal stability of the materials by thermogravimetric analysis in a nitrogen stream at a heating rate of 10 °C min^−1^. Differential scanning calorimetry (DSC) was performed on a Mettler-Toledo DSC1 module in a nitrogen atmosphere with a heating/cooling rate of 10 °C min^−1^. Indium was used as a standard for the temperature and enthalpy calibrations. A 3-angle static light scattering (MALS) detector was used to conduct the size exclusion chromatography-multi angle light scattering (SEC-MALS) measurements (i.e., miniDAWN TREOS, from Wyatt Technology). Both the detector and an Agilent 1260 infinity high performance liquid chromatography (HPLC) system (vide DMA-SEC) are coupled on-line and are used for the determination of absolute molar mass of the polymer samples. All measurements were performed under ambient conditions. The refractive index (RI) increment (dn/dc) values were estimated using online size-exclusion chromatography (SEC) coupled with a RI detector. The RI detector measures an increase for a 1–10 mg/mL concentration series of the examined polymer. Astra 7 software (Wyatt Technology) was used to analyze the obtained light scattering (LS) results.

### 2.3. Synthesis of Poly(2-Isopropenyl-2-Oxazoline) (PiPOx)

PiPOx (DP = 150) was prepared via living anionic polymerization using the optimized protocols reported in our previous papers [44,47]. In a typical run, the anionic polymerization of iPOx and all manipulations were performed in clean/dry glassware and under Ar flow. A 1.6 M solution of 0.5 g (4.77 mmol) of iPOx in 2.5 mL of THF was cooled down to −40 °C. Then, 12.7 μL of n-BuLi (0.0318 mmol) in hexane was injected into the reaction flask. The polymerization was kept for 10 min at this temperature and terminated with 1 mL of methanol. The reaction mixture was allowed to warm up to room temperature, then was diluted with 1 mL of methanol, precipitated from diethyl ether twice, and dried in a vacuum oven at 55 °C. The PiPOx polymer was obtained with a 95% yield as a fine white powder. The absolute number average molecular weight (M_n_) of the analyzed polymer was 17.2 kg mol^−1^, with a dispersity of 1.12 as determined by SEC-MALS. The specific refractive index increment value was determined (dn/dc = 0.0902 mL g^−1^). The ^1^H NMR of PiPOx is presented in Appendix A.

### 2.4. PiPOx Molecularly Imprinted Polymers (MIPs) Synthesis

PiPOx (0.3 g, 2.7 mmol of 2-oxazoline groups), DTDPA (0.278 g, 1.33 mmol), and 5-FU (0.1 g, 0.77 mmol) were dissolved in *N*,*N*’-dimethylformamide (DMF) (1.8 mL) in a pressure vial under an inert atmosphere. The vial was placed in a heating block at 80 °C for 24 h, which allowed for a full cross-linking process to be conducted. Subsequently, the obtained solid polymer (PiPOx-MIP) was ground to a fine powder, which was sieved using a 40-mesh sieve. As obtained fine powder was washed with a small amount of DMF, placed in a dialysis bag, and dialyzed against MeOH/NH_3_ (aq) (95:5) and with pure methanol, twice each. This allowed us to completely remove the template molecules from the obtained MIPs’ structure. The final polymer was placed in a vacuum oven and dried at 40 °C for 12 h to remove the residual solvent.

The non-imprinted polymer (PiPOx-NIP) was obtained using the same procedure but without adding the 5-FU template.

### 2.5. Adsorption Experiments

5-FU adsorption by PiPOx-MIP and PiPOx-NIP was investigated using batch experiments. To evaluate the adsorption kinetics, 50 mg of PiPOx-MIP or PiPOx-NIP and 50 mL of 5-FU solution at the initial concentration of 0.2 mg mL^−1^ were stirred at room temperature. At specified time intervals, the 5-FU concentration was measured using UV–Vis spectrophotometer. The amount of drug adsorbed at time *t* (*q_t_*; mg g^−1^) was calculated using the following equation:(1)$$q_{t}=\frac{\left(C_{0}-C_{t}\right)V}{m}$$
where *C_t_* is the 5-FU concentration at time *t* (h).

Adsorption isotherms were obtained by preparing a series of samples containing 20 mg of PiPOx-MIP or PiPOx-NIP mixed with 10 mL of DMF solution containing various concentrations of 5-FU (0.01–2 mg mL^−1^). All samples were allowed to equilibrate for 48 h. The 5-FU concentration was measured before and after adsorption using the UV–Vis spectrophotometer. The amount of 5-FU adsorbed (*q_eq_*; mg g^−1^) was calculated using the following equation:(2)$$q_{eq}=\frac{\left(C_{0}-C_{eq}\right)V}{m}$$
where *C*_0_ and *C_eq_* are the initial and equilibrium concentrations (mg mL^−1^); *m* is the mass of the polymer (g); and *V* is the solution volume (mL). All measurements were repeated three times and average values were used for calculations.

### 2.6. In Vitro 5-Fluorouracil (5-FU) Release Studies

Unloaded PiPOx-MIP and PiPOx-NIP were immersed for 24 h in 1 mg mL^−1^ 5-FU DMF solution to ensure maximum drug rebinding in the materials’ structure. As obtained drug-loaded materials were used in subsequent release studies. The amount of drug bonded was calculated based on initial and equilibrium 5-FU concentration in the applied solution. In vitro release studies were conducted by immersing 20 mg of 5-FU loaded PiPOx-MIP or PiPOx-NIP in phosphate-buffered saline (PBS) with pH 7.4, PBS with pH 6.5, and HCl-KCl buffer (simulated gastrointestinal conditions) with pH 2.0 and stirred at 37 °C. A second set of samples was prepared using the same conditions, but additionally, TCEP (60 mg) was added to them. For all samples, the concentration of 5-FU was measured at specified time intervals using a UV–Vis spectrophotometer. Total amounts of 5-FU released (*F_t_*; mg) were calculated using the following equation:(3)$$F_{t}=V_{m}C_{t}$$
where *V_m_* (mL) and *C_t_* (mg mL^−1^) are the volume and 5-FU concentration at time t.

The 5-FU release data were fitted to different kinetic models. Zero-order (Equation (4)), first-order (Equation (5)), Higuchi (Equation (6)), and Hixson-Crowell (Equation (7)) mathematical models were used to analyze the release data.
(4)$$F_{t}=k_{0}t$$
(5)$$F_{t}=1-e^{-kt}$$
(6)$$F_{t}=k_{H}\sqrt{t}$$
(7)$$\sqrt[{3}]{F_{0}}-\sqrt[{3}]{F_{t}}=k_{HC}t$$
where *F_t_* is the fraction of 5-FU released at time *t*; *F*_0_ is the initial amount of drug in the polymer; and *k*_0_, *k*_1_, *k_H_*, and *k_HC_* are the release constant of the respective kinetic models.

### 2.7. Statistical Analysis

One-way analysis of variance (ANOVA) with the post-hoc Tukey honest significant difference (HSD) test was used to test the statistical significance. The *p* value lower than 0.05 was considered as a statistical significance.

## 3. Results and Discussion

### 3.1. Polymer Synthesis and Characterization

For the synthesis of the MIPs, a PiPOx precursor with a M_n_ of 17.2 kDa and a dispersity of 1.12 as determined by SEC-MALS was used (Figure 1). To obtain reduction-responsive PiPOx-MIPs, a cross-linker containing a disulfide bond and terminal carboxylic groups was used. The amide groups formed during the cross-linking reaction are anticipated to interact with the 5-FU molecules, while the disulfide bond can be cleaved under reductive stress, which is achieved during drug release experiments by the addition of TCEP. The amount of DTDPA used for cross-linking was calculated to react 98% of the available 2-oxazoline groups in the PiPOx structure, which assures the formation of an almost completely cross-linked MIP.

The FTIR absorbance spectra of drug-loaded PiPOx-MIP, drug-unloaded PiPOx-MIP, and PiPOx-NIP are presented in Figure 2. All three obtained polymers showed similar characteristic bands in the IR spectra originating from the main polymer structure. In all spectra, absorption bands were visible at 3404 cm^−1^ (ν N–H, broad band) and 2936 cm^−1^ (ν C–H), 1722 cm^−1^ (ν C=O), corresponding to the ester groups, 1520 cm^−1^ (ν N–H), and at 1178 cm^−1^ (ν C–O). The adsorption bands of the amide groups (ν C=O) can be observed at 1644 cm^−1^ for drug-unloaded PiPOx-MIP and PiPOx-NIP, whereas for drug-loaded PiPOx-MIP this band is shifted to 1650 cm^−1^ and has much higher intensity in comparison to other signals. Moreover, the drug-loaded PiPOx-MIP showed additional signals at 1384 cm^−1^ and 660 cm^−1^. All these results clearly confirm the presence of 5-FU [52] in the loaded PiPOx-MIP structure and indicate that the amide groups form hydrogen bonding with 5-FU.

The SEM images of the 5-FU loaded PiPOx-MIP, unloaded PiPOx-MIP, and PiPOx-NIP are presented in Figure 3. All obtained materials showed a uniform surface with a relatively smooth structure, which is probably a result of the very high degree of cross-linking. No porosity can be seen, which means that the adsorption of 5-FU will mostly occur in the designed cavities for PiPOx-MIP and by non-selective interactions for PiPOx-NIP. No visible differences can be observed in the morphology of 5-FU loaded PiPOx-MIP, unloaded PiPOx-MIP, and PiPOx-NIP. The size of the obtained particles appeared to be smaller than the mesh sieve used (425 µm), which means that the polymer can be easily ground to reach a particular size.

The thermogravimetric analysis (TGA) of the loaded PiPOx-MIP, unloaded PiPOx-MIP, and PiPOx-NIP is shown in Figure 4a. For all of the obtained materials, a single decomposition step was observed, as the main structure of these materials was identical. The degradation started at around 250 °C and ended at around 450 °C and can be attributed to the complete decomposition of the polymer structure. For the 5-FU loaded PiPOx-MIP, the thermal degradation started at around 230 °C and a clearly visible higher weight loss in the initial phase of this step could be observed, which can be ascribed to the presence of 5-FU in the PiPOx-MIP cavities, as 5-FU undergoes complete decomposition at 285 °C [53]. The differential scanning calorimetry (DSC) results of 5-FU loaded PiPOx-MIP and PiPOx-NIP are shown in Figure 4b. DSC analysis of PiPOx-NIP revealed a broad endothermic peak around 100 °C, indicating the presence of a crystalline phase due to the newly formed ester amide bonds. The absence of the melting point around 280 °C in the DSC curve of 5-FU loaded PiPOx-MIP proved the uniform amorphous distribution of 5-FU inside the MIPs’ structure.

### 3.2. Adsorption Kinetics

Examination of the relationship between adsorption capacity and contact time of the polymers with a solution of 5-FU allowed us to establish the adsorption kinetics for PiPOx-MIP and PiPOx-NIP. The obtained plots showing the relationship between *q_t_* and *t* are presented in Figure 5. Two adsorption kinetic models, the pseudo-first-order model given by Lagergren and Svenska and the pseudo-second-order model based on the equilibrium adsorption, were used to characterize the obtained experimental data. The pseudo-first-order model is described using the following equation:(8)$$\log\left(q_{e}-q_{t}\right)=\log q_{e}-\frac{k_{1}}{2.303}t$$
where *k*_1_ (h^−1^) is the pseudo-first-order rate constant; *q_e_* (mg g^−1^) is the amount of 5-FU adsorbed at equilibrium concentration; and *q_t_* (mg g^−1^) is the amount of 5-FU adsorbed at time *t* (h). The calculated *k*_1_ and R^2^ values are given in Table 1. The correlation coefficient (R^2^) values obtained were equal to 0.963 and 0.994 for PiPOx-MIP and PiPOx-NIP, respectively. The very high R^2^ value obtained for PiPOx-NIP clearly suggests that the pseudo-first-order model can be used to describe the kinetics of adsorption of 5-FU on this polymer. However, the R^2^ value obtained for PiPOx-MIP was moderately low, especially in comparison with the R^2^ parameter obtained for the pseudo-second-order kinetic model. Consequently, the pseudo-first-order model should not be used to characterize the adsorption kinetics of 5-FU on the PiPOx-MIP. Moreover, the fact that the adsorption kinetic process for both adsorbents is described using two different models accounts for two different mechanisms that are responsible for the adsorption. For PiPOx-MIP, it may be postulated that the adsorption occurs within the cavities formed during imprinting, whereas for PiPOx-NIP, it is non-selective surface adsorption.

The pseudo-second-order model is described using the following equation:(9)$$\frac{t}{q_{t}}=\frac{1}{k_{2}q_{e}^{2}}+\frac{1}{q_{e}}t$$
where *k*_2_ (g mg^−1^ h^−1^) is the pseudo-second-order rate constant. The calculated *k*_1_ and R^2^ values are given in Table 1. The R^2^ values were equal to 0.984 and 0.900 for PiPOx-MIP and PiPOx-NIP, respectively. The results clearly indicate that the pseudo-second-order kinetic model should only be used to characterize the adsorption kinetic of 5-FU on PiPOx-MIP. The values of the kinetics constants, *k*_1_ and *k*_2_, were both higher for PiPOx-MIP, which suggests that adsorption occurs faster on this material.

### 3.3. Adsorption Isotherms

The adsorption process was investigated by determining adsorption isotherms for the 5-FU adsorption at the equilibrium state. The relationship between the 5-FU equilibrium concentration and the amount of adsorbed 5-FU by the corresponding polymer is presented in Figure 6. Two different models, namely the Langmuir and Freundlich adsorption isotherm models, were used for data interpretation. The Langmuir adsorption isotherm is described using the following equation:(10)$$\frac{C_{eq}}{q_{eq}}=\frac{C_{eq}}{q_{m}}+\frac{1}{Kq_{m}}$$
where *K* (L mg^−1^) is the binding equilibrium constant; *q_m_* (mg g^−1^) is the maximum amount of 5-FU adsorbed; *C_eq_* (mg L^−1^) is the equilibrium concentration of 5-FU; and *q_eq_* (mg g^−1^) is the amount of 5-FU adsorbed at equilibrium concentration. The calculated *K*, *q_m_*, and R^2^ (correlation coefficients) values are summarized in Table 2. The R^2^ values obtained for both PiPOx-MIP and PiPOx-NIP were higher than 0.99, which clearly indicates that the Langmuir adsorption model very nicely fit the experimental data. The value of maximum adsorption capacity (*q_m_*) for the PiPOx-MIP was equal to 85.3 mg g^−1^, whereas for the PiPOx-NIP, it was equal to 66.1 mg g^−1^, proving that the imprinting was successful. Moreover, it can be concluded that non-selective adsorption is also significant, as the imprinting process increases the adsorption capacity by around 30%. This observation can be explained by the presence of the ester-amide groups in the cross-linked PiPOx, after the 2-oxazoline ring is opened, which can interact with the 5-FU.

The second model that was applied was the Freundlich adsorption isotherm, which can be described by the following equation:(11)$$q_{eq}=K_{f}C_{eq}^{1/n}$$
(12)$$\log q_{eq}=\log K_{f}+\frac{1}{n}\log C_{eq}$$
where *K_f_* and *n* represent the Freundlich constants; *C_eq_* (mg L^−1^) is the equilibrium concentration of 5-FU; and *q_eq_* (mg g^−1^) is the amount of 5-FU adsorbed at equilibrium concentration. The calculated *K_f_*, *1/n*, and R^2^ (correlation coefficients) values are summarized in Table 2. The R^2^ values obtained for PiPOx-MIP and PiPOx-NIP were equal to 0.910 and 0.919, respectively, suggesting that the Freundlich adsorption model only partially fit the obtained experimental data and the fitting was less accurate than for the Langmuir model. In the Freundlich adsorption model, the *1/n* parameter is considered as a measure of adsorption intensity or surface heterogeneity. When the value of this parameter gets closer to zero, it means that the surface is more heterogeneous [54]. In contrast, when the value of this parameter is higher than one, it shows that adsorption is cooperative [55]. For both polymers, the values were very similar and were equal to 0.53 and 0.59 for PiPOx-MIP and PiPOx-NIP, respectively, indicating that the heterogeneity of both materials is quite similar.

### 3.4. In Vitro Release Studies

In vitro release experiments of 5-FU were performed in three buffer solutions simulating gastric fluid (pH 2.0), tumor interstitium of tumor cells (pH 6.5), and intravenous conditions (pH 7.4). The drug loading on the basis of rebinding experiments was estimated to be 78 mg g^−1^ for PiPOx-MIP and 59 mg g^−1^ for PiPOx-NIP. The release experiments were performed in the absence and with the addition of TCEP to investigate the reduction-responsiveness of the obtained PiPOx-MIP. Moreover, control experiments were performed with the use of PiPOx-NIP to examine whether the increased release observed for PiPOx-MIP in the presence of TCEP was related to MIP cavity degradation or degradation of the entire network structure.

The release profiles of 5-FU from both PiPOx-MIP and PiPOx-NIP with and without the addition of TCEP are presented in Figure 7. The addition of TCEP significantly influenced the release of 5-FU from PiPOx-MIP (*p* < 0.05), whereas for PiPOx-NIP, no statistically important differences in 5-FU release were observed with and without TCEP. Thus, one may conclude that TCEP causes changes of the PiPOx-MIP structure by cleaving disulfide bonds, which results in the degradation of the cavities responsible for 5-FU adsorption. The highest cumulative release was observed for PiPOx-MIP after the addition of TCEP at pH 7.4 (ca. 58%), a slightly lower release at pH 6.5 (ca. 55%), and an even smaller release at pH 2.0 (ca. 46%). Hence, the increase in the pH led to an increased drug release, although the differences in the cumulative release were not that significant. The observed limited release was probably caused by using different conditions for 5-FU reloading and release. Reloading was done in DMF to ensure a high loading ratio as this solvent is used during the synthesis of imprinted material. The release was done in aqueous solutions, which might be a cause of limited release.

The release profile was similar for all of the examined pH values, starting with an initial burst release during the initial ca. 20 min, followed by a steady release that lasted for 4 to 5 h. The release profiles and pH dependence for PiPOx-MIP in the presence and absence of TCEP were almost identical, despite a higher cumulative release being achieved in the presence of TCEP. The initial burst release was similar and dominating for all materials, with and without TCEP, indicating that it is a diffusion controlled release of readily accessible 5-FU. The TCEP influences the maximum release after this initial burst release process finishes as the slower (compared to initial burst release) cleavage of the disulfide bond allows further release of less accessible 5-FU. These results demonstrate that our system does not function as a pH controlled release system while it has potential as a reduction-triggered release system.

For the 5-FU release from PiPOx-NIP, no differences were observed, regardless of whether TCEP was added or not to the release medium, suggesting that TCEP causes disulfide bond cleavage, which affects the cavities formed during PiPOx-MIP synthesis while it does not influence the physisorption responsible for 5-FU adsorption by PiPOx-NIP. Moreover, no significant differences were noticed for the release of 5-FU from PiPOx-NIP under various pH conditions, as for all three examined pH values, the highest cumulative release reached ca. 33%.

The proposed reduction-responsive system presents a good alternative for already developed pH responsive 5-FU N-succinyl chitosan/poly(acrylamide-*co*-acrylic acid) hydrogel-based systems [56]. These pH-responsive systems show relatively high cumulative 5-FU release (60–80%) at pH 7.4 and low 5-FU release at low pH (10–17% at pH 1.2). The here presented PiPOx-MIP-based system shows a moderate release at pH 2.0 (ca. 46%), and a similar release to the chitosan/poly(acrylamide-*co*-acrylic acid) hydrogel-based system and pH 7.4. When compared with other reduction-responsive systems, the TCEP-induced release in the presented system showed a higher percentage increase in overall 5-FU release. For cystamine conjugated chitosan-SS-mPEG, it was observed that 5-FU release without the triggering agent (glutathione) reached 60%, while after its addition, the release was increased by ca. 18% [57]. For the PiPOx-MIP-based system, the TCEP addition increased release by ca. 28%, 20%, and 22% at pH 7.4, 6.5, and 2.0, respectively.

Four mathematical models were applied to fit the experimental data obtained for 5-FU release from PiPOx-MIP and PiPOx-NIP, both with and without the addition of TCEP. Zero-order, first-order, Higuchi, and Hixson-Crowell release models were used to fit the release profiles. Table 3 summarizes the obtained values for the corresponding release constants (*k*), correlation coefficients (R^2^), and diffusion exponent (*n*). For all examined materials in all released media, the highest R^2^ values were obtained for the Higuchi model. Nonetheless, the release profiles of the 5-FU were only nicely fitted for PiPOx-MIPs at pH 6.5 (R^2^ > 0.9), while a rather low correlation coefficient (0.6 < R^2^ < 0.9) could be found in all the other cases, indicating an interplay of multiple release mechanisms. These similar results for the different polymers indicate that the 5-FU release mechanism for all materials and all release media is similar, which might be caused by an almost identical structure of the polymers. The 5-FU entrapped within the PiPOx-MIP cavities still needs to pass the PiPOx-MIP surface, during which it encounters similar interactions as for the non-imprinted polymer. Presumably, the interactions of 5-FU with the PiPOx surface are strong enough to dominate the final drug release mechanism.

## 4. Conclusions

In conclusion, we have shown that the synthesis of molecularly imprinted PiPOx with 5-FU can be easily performed using PiPOx and dicarboxylic acid. The imprinted polymer materials showed a higher adsorption capacity than the not imprinted analogs, demonstrating that the imprinting was successful. Adsorption kinetic data proved that the mechanism of 5-FU adsorption is different for PiPOx-MIP and PiPOx-NIP. The addition of TCEP presumably caused degradation of the PiPOx-MIP cavities, which resulted in increased drug release. The highest cumulative release for PiPOx-MIP after TCEP addition was obtained for pH 7.4 and the lowest for pH 2.0. The highest cumulative release for the same polymer, but without TCEP addition, was significantly lower than when TCEP was added. For the release of 5-FU from PiPOx-NIP, no statistically significant differences were observed whether TCEP was added or not. The 5-FU release from all examined materials at all pH values was well fitted with the Higuchi model. Overall, we have demonstrated that the molecularly imprinted PiPOx possessing disulfide units in its structure can be efficiently used for the reduction-triggered responsive release of 5-FU. The current size of the obtained particles would allow them to be used in localized anticancer therapy through injection into the tumor site. Further research is needed to develop a system for intravenous administration for which nanometer sized particles will be required.

## Figures and Tables

**Figure 1 pharmaceutics-12-00506-f001:**
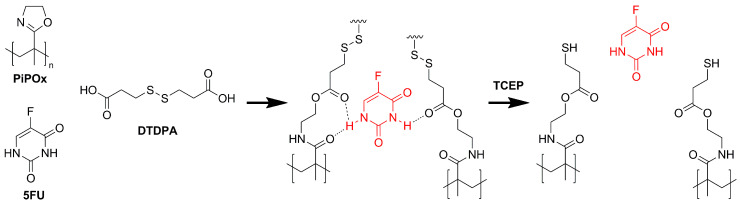
Schematic representation of the poly(2-isopropenyl-2-oxazoline) molecularly imprinted polymer (PiPOx-MIP) with 5-fluorouracil (5-FU): synthesis and subsequent reduction-triggered 5-FU release with tris(2-carboxyethyl)phosphine (TCEP).

**Figure 2 pharmaceutics-12-00506-f002:**
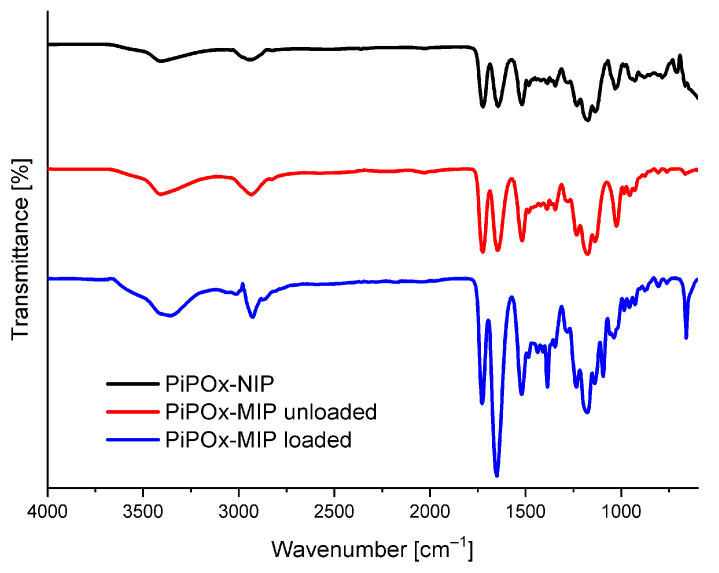
Fourier transform infrared spectroscopy (FTIR) spectra of drug-loaded PiPOx-MIP, drug-unloaded PiPOx-MIP, and PiPOx- non-imprinted polymers (NIP).

**Figure 3 pharmaceutics-12-00506-f003:**
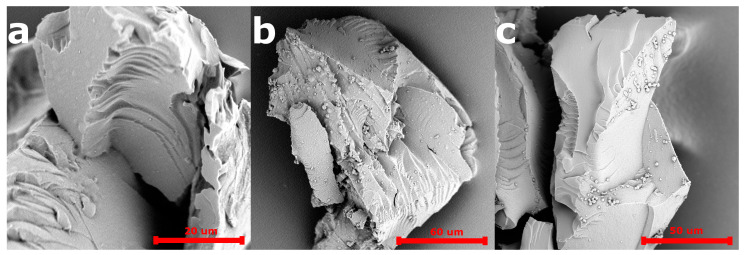
Scanning electron microscopy (SEM) images of PiPOx-NIP (**a**), unloaded PiPOx-MIP (**b**), and loaded PiPOx-MIP (**c**).

**Figure 4 pharmaceutics-12-00506-f004:**
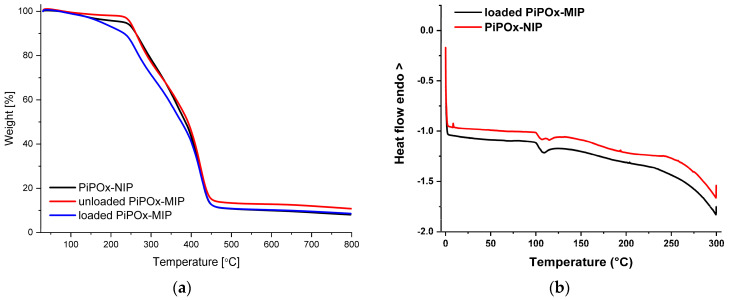
(**a**) Thermogravimetric analysis (TGA) curves of loaded PiPOx-MIP, unloaded PiPOx-MIP, and PiPOx-NIP; (**b**) Differential scanning calorimetry (DSC) curves for the fourth heating of the loaded PiPOx-MIP and PiPOx-NIP.

**Figure 5 pharmaceutics-12-00506-f005:**
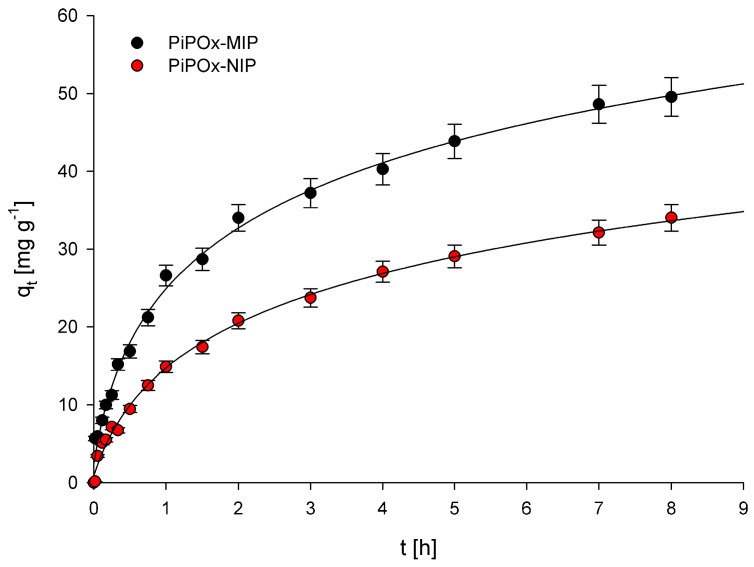
Plots of q_t_ versus time obtained for PiPOx-MIP and PiPOx-NIP.

**Figure 6 pharmaceutics-12-00506-f006:**
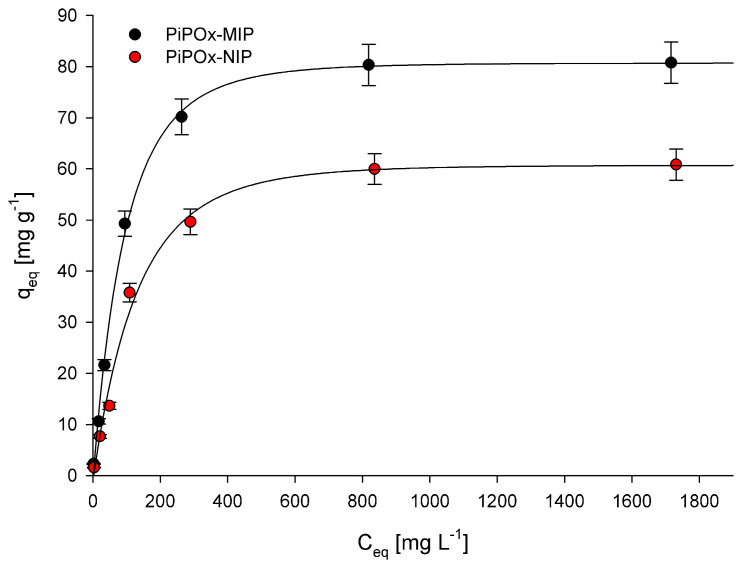
Adsorption isotherms of 5-FU in PiPOx-MIP and PiPOx-NIP.

**Figure 7 pharmaceutics-12-00506-f007:**
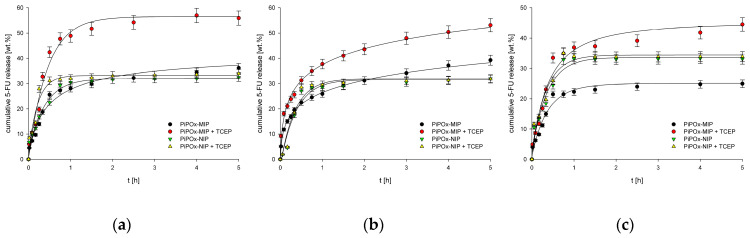
(**a**) Release profiles of 5-FU from drug loaded PiPOx-MIPs and NIPs at pH 7.4; (**b**) Release profiles of 5-FU from drug loaded PiPOx-MIPs and NIPs at pH 6.5; (**c**) Release profiles of 5-FU from drug loaded PiPOx-MIPs and NIPs at pH 2.0.

**Table 1 pharmaceutics-12-00506-t001:** Kinetic parameters calculated for the pseudo-first-order and pseudo-second-order models.

Polymer	Pseudo-First-Order Kinetic Model		Pseudo-Second-Order Kinetic Model	
	k_1_ [h^−1^]	R^2^	k_2_ [g mg^−1^ h^−1^]	R^2^
PiPOx-MIP	0.47 ± 0.08	0.963	0.022 ± 0.004	0.984
PiPOx-NIP	0.38 ± 0.06	0.994	0.015 ± 0.003	0.900

**Table 2 pharmaceutics-12-00506-t002:** Parameters of 5-FU adsorption by PiPOx-MIP and PiPOx-NIP.

Polymer	Langmuir			Freundlich		
	q_m_ [mg g^−1^]	K [L mg^−1^]	R^2^	K_f_ [mg g^−1^ (L mg^−1^)^1/n^]	1/n	R^2^
PiPOx-MIP	85.3 ± 1.6	12.4 ± 0.4	0.998	102 ± 6	0.53 ± 0.08	0.910
PiPOx-NIP	66.1 ± 1.3	7.94 ± 0.36	0.996	74 ± 5	0.59 ± 0.07	0.919

**Table 3 pharmaceutics-12-00506-t003:** Release kinetic data of 5-FU from PiPOx-MIP and PiPOx-NIP.

Polymer	Medium	Zero Order		First Order		Higuchi		Hixson-Crowell	
		k_0_ [h^−1^]	R^2^	k_1_ [h^−1^]	R^2^	k_H_ [h^−1/2^]	R^2^	k_HC_ [h^−1/3^]	R^2^
PiPOx-MIP	pH 7.4	2.73 ± 0.29	0.676	0.123 ± 0.014	0.471	10.4 ± 1.6	0.838	0.112 ± 0.013	0.543
	pH 7.4 + TCEP	3.53 ± 0.37	0.491	0.114 ± 0.013	0.361	14.2 ± 1.5	0.677	0.116 ± 0.013	0.406
	pH 6.5	4.11 ± 0.43	0.770	0.175 ± 0.019	0.530	12.7 ± 1.3	0.924	0.163 ± 0.018	0.622
	pH 6.5 + TCEP	5.43 ± 0.56	0.765	0.162 ± 0.018	0.567	16.9 ± 1.8	0.922	0.171 ± 0.019	0.641
	pH 2.0	1.41 ± 0.15	0.475	0.094 ± 0.010	0.363	5.72 ± 0.59	0.659	0.075 ± 0.009	0.402
	pH 2.0 + TCEP	2.92 ± 0.30	0.569	0.116 ± 0.013	0.401	11.5 ± 1.6	0.747	0.110 ± 0.012	0.459
PiPOx-NIP	pH 7.4	4.75 ± 0.49	0.545	0.256 ± 0.027	0.460	13.0 ± 1.3	0.785	0.222 ± 0.024	0.491
	pH 7.4 + TCEP	5.71 ± 0.59	0.440	0.305 ± 0.032	0.375	14.5 ± 1.5	0.690	0.265 ± 0.028	0.399
	pH 6.5	3.57 ± 0.37	0.437	0.283 ± 0.029	0.308	11.9 ± 1.3	0.661	0.205 ± 0.022	0.350
	pH 6.5 + TCEP	3.39 ± 0.35	0.390	0.267 ± 0.028	0.295	11.6 ± 1.3	0.617	0.195 ± 0.021	0.327
	pH 2.0	4.20 ± 0.43	0.454	0.211 ± 0.023	0.396	12.3 ± 1.4	0.702	0.187 ± 0.019	0.419
	pH 2.0 + TCEP	4.07 ± 0.41	0.430	0.192 ± 0.020	0.386	12.0 ± 1.4	0.677	0.176 ± 0.018	0.404

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
