# Peer review of "Reduction-Responsive Molecularly Imprinted Poly(2-isopropenyl-2-oxazoline) for Controlled Release of Anticancer Agents"

_pharmaceutics, 2020, doi:10.3390/pharmaceutics12060506_

Round 1

Reviewer 1 Report

This is a well written manuscript.

However, some information should be presented to improve the article.

  1. What is the 5-FU percentage loaded in PiPOx-MIP? Is it possible to detect the unreacted drug? Was it performed?
  2. Do you have any ideia about the toxicity of the 5-FU-imprinted PiPOx on cancer cells and on non-cancer cells?
  3. To enrich the manuscript, the design of the new particles should be included.

Author Response

  1. What is the 5-FU percentage loaded in PiPOx-MIP? Is it possible to detect the unreacted drug? Was it performed?

PiPOx-MIP after synthesis was dialyzed to remove all 5-FU, therefore the content of drug in this material after synthesis was not studied.  For the release studies reloading of 5-FU was applied. The additional data has been added to the manuscript regarding this process. The following text was added: “Unloaded PiPOx-MIP and PiPOx-NIP were immersed for 24 h in 1 mg mL-1 5-FU solution to ensure maximum drug adsorption in the materials. These drug-loaded materials were used in subsequent release studies. The amount of adsorbed drug was calculated based on initial and equilibrium 5-FU concentration in the applied solution.” The percentage of drug loading using this method is 8% (m/m) for PiPOx-MIP and 6% (m/m) for PiPOx-NIP.

  1. Do you have any ideia about the toxicity of the 5-FU-imprinted PiPOx on cancer cells and on non-cancer cells?

This data was not studied in the research, as we focused mostly on synthesis of reduction-responsive materials with 5-FU as a proof of concept drug. The experiments proposed by the Reviewer are of high interest and will probably be studied in the future. Nonetheless, the PiPOx base polymer has previously been shown to be harmless for cells, both as soluble polymer and as hydrogel (Macromol. Biosci. 2016, 16, 1200-1211 & Chem. Mater. 2018, 30, 7938-7949).

  1. To enrich the manuscript, the design of the new particles should be included.

The structural design of the particles is presented in Figure 1, which also includes the proposed interactions occurring between 5-FU and MIPs cavities. When it comes to morphology of the polymer particles the following sentence was added: “The size of obtained particles appears to be smaller than the mesh sieve used (425 µm), which means that the polymer can be easily grounded to reach particular size.”.

Reviewer 2 Report

The manuscript presents the synthesis and physico-chemical characterization of molecularly imprinted polymers for reduction-responsive 5-fluorouracil release against different pHs. The study is well conducted, organized. All results demonstrated the successful polymer synthesis. However, minor concerns are related to the improvement of the discussion section, since results are presented but lack of literature comparisons, especially regarding to the delivery performance against other stimuli-sensitive systems for 5-fluorouracil delivery.

Author Response

The manuscript presents the synthesis and physico-chemical characterization of molecularly imprinted polymers for reduction-responsive 5-fluorouracil release against different pHs. The study is well conducted, organized. All results demonstrated the successful polymer synthesis. However, minor concerns are related to the improvement of the discussion section, since results are presented but lack of literature comparisons, especially regarding to the delivery performance against other stimuli-sensitive systems for 5-fluorouracil delivery.

We would like to thank Reviewer for nice comments. The discussion including literature comparison with other stimuli-senstive systems for 5-FU delivery was added as separate paragraph in section 3.4.

Reviewer 3 Report

The subject of the reviewed publication is the method of synthesis and properties of imprinted poly (2-isopropenyl-2-oxazoline) for controlled release of anticancer agent  (5-fluorouracile). Authors  used the properties of molecularly imprinted polymers structures in controlled release of the fluorouracil. Molecular imprinting is a rapidly developing technique for the preparation of polymeric materials that are capable of molecular recognition for selective separation and chemical identification. The authors have already successfully used poly (2-oxazoline) as based MIPS structure, with indomethacin as the template. Currently, the synthesis of poly (2-oxazoline)s and derivative materials is well known and widely described in literature [e.g. Polymers 2013, 5, 956-1011; doi: 10.3390 / polym5030956], as well as the use of this polymer in drug delivery processes [Bioconjugate Chem. 2011, 22, 5, 976-986]. However, we must acknowledge the authors that 5-fluorouracile has not been previously used as a template in forming MIPS based on this type of polymers. However the presented idea is not a great novelty, because this cytostatics has long been used in MIPS systems - also with similar biodegradable hydrogels such as; poly derivatives (2-hydroxyethylmetacrylate-cl-acrylic acide) [e.g. Acta Biomaterialia, 4 (2008) 1244–1254.] or copolymers of ethylene glycol dimethacrylate (EGDMA) with methacrylic acid   [Molecules 2007, 12 (4), 805-814]. Unfortunately, in the introduction part of the manuscript the authors mention nothing about these works , what they should to do.

The novelty of this work seems to be the introduction of disulfide bond into the network of a synthesized  gel, through a convenient route using dithiodipropionic acide (DTDPA). The idea of introducing disulfide bonds into derivative poly (oxazoline) has been known for a long time [Macromolecules 1993, 26, 5, 883-887]. The reduction-sensitive disulfide bond is usually employed as a linkage between polymers and drugs, or as cross-linkers in polymeric drug carriers. These polymeric drug carriers can release of anticancer drug in response to the reducing or enzymatic environment when they reach tumor tissues [Medicinal Research Reviews, 17 Jan 2018, 38 (5): 1485-1510].

Although many of the elements presented in the presented work are largely imitative, it also has many potentially interesting and worth elements. Belong to them the research on the kinetics of fluorouracil absorption and its release in vitro. Unfortunately, the presented results raise great uncertainty as to their reliability.  This is mainly due to doubts about properly measured changes in drug concentration. The observed shape of some of the obtained release profiles may be sign the  such irregularities. There is practically no description of the conditions and methodology for conducting UVVIS measurements. As presented in earlier works, the results reliability of such measurement are influenced by many factors including the choice of the wavelength and selection of concentrations and measurement conditions in creating the calibration graph, especially when measurements are carried out at different pH  [NanoBiomed Eng. 2018, 10 (3): 224-234.]

Unfortunately, the obtained results of investigations was not compared to the similar results obtained earlier by other, showing the absorption and release of fluoroacil from other MiPS. The creation of a new material does not mean that it is better, it should be presented against the background of other, earlier ones. There is lack of  the scientific summary of the  obtained results.

The assessed work is not suitable for publication in present form. Requires many corrections and additions.

Author Response

The subject of the reviewed publication is the method of synthesis and properties of imprinted poly (2-isopropenyl-2-oxazoline) for controlled release of anticancer agent  (5-fluorouracile). Authors  used the properties of molecularly imprinted polymers structures in controlled release of the fluorouracil. Molecular imprinting is a rapidly developing technique for the preparation of polymeric materials that are capable of molecular recognition for selective separation and chemical identification. The authors have already successfully used poly (2-oxazoline) as based MIPS structure, with indomethacin as the template. Currently, the synthesis of poly (2-oxazoline)s and derivative materials is well known and widely described in literature [e.g. Polymers 2013, 5, 956-1011; doi: 10.3390 / polym5030956], as well as the use of this polymer in drug delivery processes [Bioconjugate Chem. 2011, 22, 5, 976-986]. However, we must acknowledge the authors that 5-fluorouracile has not been previously used as a template in forming MIPS based on this type of polymers. However the presented idea is not a great novelty, because this cytostatics has long been used in MIPS systems - also with similar biodegradable hydrogels such as; poly derivatives (2-hydroxyethylmetacrylate-cl-acrylic acide) [e.g. Acta Biomaterialia, 4 (2008) 1244–1254.] or copolymers of ethylene glycol dimethacrylate (EGDMA) with methacrylic acid   [Molecules 2007, 12 (4), 805-814]. Unfortunately, in the introduction part of the manuscript the authors mention nothing about these works , what they should to do.

We thank the Reviewer for critically assessing our manuscript. However, we would like to point out that the reported system is a highly crosslinked MIP that has very different properties compared to the hydrogel systems that are referred to by the Reviewer. Nonetheless, we agree that the introduction should be expanded with information suggested by the Reviewer. We would like to point out that this is only the second report regarding poly(oxazoline)-based MIPs, while being the first on a PiPrOx-based MIP, which is crosslinked via ring-opening of the side chain 2-oxazoline rings. Therefore, in our view this topic is quite novel.

The novelty of this work seems to be the introduction of disulfide bond into the network of a synthesized  gel, through a convenient route using dithiodipropionic acide (DTDPA). The idea of introducing disulfide bonds into derivative poly (oxazoline) has been known for a long time [Macromolecules 1993, 26, 5, 883-887]. The reduction-sensitive disulfide bond is usually employed as a linkage between polymers and drugs, or as cross-linkers in polymeric drug carriers. These polymeric drug carriers can release of anticancer drug in response to the reducing or enzymatic environment when they reach tumor tissues [Medicinal Research Reviews, 17 Jan 2018, 38 (5): 1485-1510].

We agree with the Reviewer that other disulfide crosslinked systems have been reported for drug release and, therefore, the introduction part was expanded with information suggested by the Reviewer. However, we would like to emphasize that the disulfide cross-linked poly(2-oxazoline) was a hydrogel system and not a MIP.

Although many of the elements presented in the presented work are largely imitative, it also has many potentially interesting and worth elements. Belong to them the research on the kinetics of fluorouracil absorption and its release in vitro. Unfortunately, the presented results raise great uncertainty as to their reliability.  This is mainly due to doubts about properly measured changes in drug concentration. The observed shape of some of the obtained release profiles may be sign the  such irregularities. There is practically no description of the conditions and methodology for conducting UVVIS measurements. As presented in earlier works, the results reliability of such measurement are influenced by many factors including the choice of the wavelength and selection of concentrations and measurement conditions in creating the calibration graph, especially when measurements are carried out at different pH  [NanoBiomed Eng. 2018, 10 (3): 224-234.]

We agree with the Reviewer that these experimental details were missing in the original submission. We have included the following text in the manuscript: “Calibration curves were plotted for all 5-FU solutions used in the study (DMF and water solutions at pH 2.0, 6.5, and 7.4) for the 0.001 – 0.05 mg mL-1 concentration range. The absorbance of the solutions was measured at 270 nm for DMF solutions and 266 nm for all water solutions. Samples containing higher concentrations than the ones used for calibration curve preparation were diluted prior to the measurement.” Similar to the research cited by the Reviewer, we have not observed any major changes in λmax of 5-FU in various solution pH. We do not think that adding graphs of calibration curve is necessary, as it will make the manuscript too long, but if the Reviewer thinks otherwise we will place this graphs as a separate figure.

Unfortunately, the obtained results of investigations was not compared to the similar results obtained earlier by other, showing the absorption and release of fluoroacil from other MiPS. The creation of a new material does not mean that it is better, it should be presented against the background of other, earlier ones. There is lack of  the scientific summary of the  obtained results.

This was also pointed out by Reviewer 2, therefore a separate paragraph including comparison with other trigger-responsive MIPs was added to the text in section 3.4

The assessed work is not suitable for publication in present form. Requires many corrections and additions.

We hope that the Reviewer will be pleased with the corrected version of the manuscript.

Reviewer 4 Report

The work is well done and the overall manuscript is well organized. The data provided is of interest and is well discussed. The authors did a good job introducing the subject and designing the experiments. This was a very light and intertaining read. Data is scientifically sound and as such acceptable for publication. 

Author Response

The work is well done and the overall manuscript is well organized. The data provided is of interest and is well discussed. The authors did a good job introducing the subject and designing the experiments. This was a very light and intertaining read. Data is scientifically sound and as such acceptable for publication. 

We are very grateful to the Reviewer for very nice comments.

Reviewer 5 Report

The paper is devoted to the obtaining of poly(2-isopropenyl-2-oxazoline) polymer cross-linked by 3,3’-dithiodipropionic acid and containing molecularly-imprinted 5-fluorouracil as a perspective formulation for reduction-responsive release. The idea of the paper is interesting and the chemistry is beautiful. However, the pharmaceutical soundness is poor and major corrections should be made before publication.

Comments:

  • Page 2, line 47. The authors wrote: “Molecularly imprinted polymers (MIPs) represent a promising class of materials due to their unique properties and can be regarded as analogs of the natural antibody-antigen systems.” Does this phrase really makes sense in the case of systems, which authors produce? There are also other words about MIP, which not help to realize the pharmaceutical interest to MIP systems developed by you. Please, specify the pharmaceutical benefits of MIP in the case of your systems and what was already done by other researches.
  • Page 2, line 86: “Within this paper, we will discuss the physicochemical properties of the obtained MIPs….” These properties should be related somehow to the pharmaceutical application of prepared formulations.
  • Page 3, line 123 “The absolute molar mass of the analyzed polymer was 17.2 kDa, with amolar mass distribution of 1.12 as determined by SEC-LS” + Page 5, line 170 “…PiPOx precursor with a molecular mass of 17.2 kDa and a molar mass distribution of 1.12 as…”. The molecular mass and molar mass are different terms and have different meaning and should be used more accurately. Molar mass could not be given in kDa, but in g/mol. Molar mass distribution seems to be wrong term. Use molecular weight distribution or dispersity Ð, which is recommended by IUPAC. Molecular mass is usually determined for polymers and it should be stated if it is Mn or Mw. Make the terminology clear. Did you applied SEC-MALS to determine absolute molecular mass? If yes, please write SEC-MALS.
  • Page 6, line 201. The shape, size and morphology of proposed formulation should be discussed as related to the proposed pharmacological application. How the system is planned to be applied to treat the tumor. Particles, films, sponges… The morphology of the polymer within the final formulation will be much more useful.
  • Page 6, line 211. It is interesting if the 5-FU forms its own crystalline phase within the polymer phase and how MIP formation could affect this. DSC or XRD measurements could be easily done for such systems to clarify this situation.
  • Page 6, Line 214 and Page 8, Line 248. It is a little bit strange that isotherm study is coming before the kinetics. Usually the kinetic studies allow finding the time at which the equilibrium state could be reached and then the isotherm could be studied based on this knowledge. I recommend the authors to follow the classical way.
  • Page 7, line 233. The imprinting factor (q-MIP/q-NIP) is around 1.3, which is quite low as compared to the systems, which are used for selective sorption. Is it enough in your case and is there any ways to improve it (if needed). If the change in cross-linking degree make sense in this case.
  • The drug adsorption was carefully studied, but the data on drug loading and encapsulation efficacy are definitely missing, as they are very important for pharmaceutical application of the system. The adsorption studies description could be minimized and finalized with the conclusion, which describe the system as pharmacologically perspective.
  • Page 9, line 285 “The addition of TCEP only significantly influences the release…” The meaning of only is unclear.
  • Page 9, line 302. There are many questions on the release data. (1) The shape of material, its form and specific surface area are unclear and don’t allow to understand the release peculiarities. As well as no data on how much of the drug was loaded. (2) If the drug has enough space to diffuse in the matrix? It seems that not, because you can reach only half of the drug released. (3) What happens with the cross-linked polymer in reductive medium? Swelling, weight loss etc effects are not described. If the reductive medium is relevant to the biological reductive strength? It is also known, that reductive medium is intracellular one. In which form the formulation can be internalized by cells? (4) The amount of the drug released is different, but the triggered release means that the rate of release is different after subjection to the trigger. If the rate is really so different? Please discuss this point more in more detailed manner. (5) It is a little bit strange that release in very acidic medium is the same as at pH 7.4. The protons are reductive themselves as well as 5-FU amide groups should be protonated and favor fast dissolution of the drug. What was the barrier for fast release in this case?
  • It is also important to give some data on the polymer toxicity. At least some from previous studies.

With best regards,

                Reviewer.

Author Response

The paper is devoted to the obtaining of poly(2-isopropenyl-2-oxazoline) polymer cross-linked by 3,3’-dithiodipropionic acid and containing molecularly-imprinted 5-fluorouracil as a perspective formulation for reduction-responsive release. The idea of the paper is interesting and the chemistry is beautiful. However, the pharmaceutical soundness is poor and major corrections should be made before publication.

Comments:

  • Page 2, line 47. The authors wrote: “Molecularly imprinted polymers (MIPs) represent a promising class of materials due to their unique properties and can be regarded as analogs of the natural antibody-antigen systems.” Does this phrase really makes sense in the case of systems, which authors produce? There are also other words about MIP, which not help to realize the pharmaceutical interest to MIP systems developed by you. Please, specify the pharmaceutical benefits of MIP in the case of your systems and what was already done by other researches.

The introduction has been updated based on the Reviewer suggestion, hereby the information on the use of MIPs not related to pharmaceutical applications has been reduced to a minimum and additional information about synthesis of MIPs-based DDS by other researchers has been added.

  • Page 2, line 86: “Within this paper, we will discuss the physicochemical properties of the obtained MIPs….” These properties should be related somehow to the pharmaceutical application of prepared formulations.

This manuscript is one of the first reports of using PiPOx in the synthesis of MIPs, therefore in our opinion as synthetic chemists, it is also necessary to physicochemical properties of the obtained materials which are not directly relevant for their pharmaceutical application.

  • Page 3, line 123 “The absolute molar mass of the analyzed polymer was 17.2 kDa, with amolar mass distribution of 1.12 as determined by SEC-LS” + Page 5, line 170 “…PiPOx precursor with a molecular mass of 17.2 kDa and a molar mass distribution of 1.12 as…”. The molecular mass and molar mass are different terms and have different meaning and should be used more accurately. Molar mass could not be given in kDa, but in g/mol. Molar mass distribution seems to be wrong term. Use molecular weight distribution or dispersity Ð, which is recommended by IUPAC. Molecular mass is usually determined for polymers and it should be stated if it is Mn or Mw. Make the terminology clear. Did you applied SEC-MALS to determine absolute molecular mass? If yes, please write SEC-MALS.

We agree with the reviewer, thus the text was modified according to the Reviewer’s suggestions.

  • Page 6, line 201. The shape, size and morphology of proposed formulation should be discussed as related to the proposed pharmacological application. How the system is planned to be applied to treat the tumor. Particles, films, sponges… The morphology of the polymer within the final formulation will be much more useful.

We agree with the Reviewer and the Conclusion section has been updated with the following information: “The current size of the obtained particles would allow them to be used in localized anticancer therapy through injection into the tumor site. Further research is needed to develop a system for intravenous administration for which nanometer sized particles will be required.”.

  • Page 6, line 211. It is interesting if the 5-FU forms its own crystalline phase within the polymer phase and how MIP formation could affect this. DSC or XRD measurements could be easily done for such systems to clarify this situation.

We agree with the Reviewer and, therefore, DSC measurements were performed and the results are now included in the manuscript. The following text was added, which answers the Reviewers question: “Differential Scanning Calorimetry (DSC) results of 5-FU loaded PiPOx-MIP and PiPOx-NIP are shown in Figure 4b. DSC analysis of PiPOx-NIP revealed a broad endothermic peak around 100 ºC indicating the presence of a crystalline phase due to the newly formed ester amide bonds. The absence of the melting point around 280 ºC in the DSC curve of 5-FU loaded PiPOx-MIP proves an uniform amorphous distribution of 5-FU inside the MIPs structure”.

  • Page 6, Line 214 and Page 8, Line 248. It is a little bit strange that isotherm study is coming before the kinetics. Usually the kinetic studies allow finding the time at which the equilibrium state could be reached and then the isotherm could be studied based on this knowledge. I recommend the authors to follow the classical way.

The order has been updated in accordance with Reviewers suggestion.

  • Page 7, line 233. The imprinting factor (q-MIP/q-NIP) is around 1.3, which is quite low as compared to the systems, which are used for selective sorption. Is it enough in your case and is there any ways to improve it (if needed). If the change in cross-linking degree make sense in this case.

From what we have observed, the high cross-linking degree in most cases ensures higher imprinting factor. This is of course to some extent, as the presence of an appropriate amount of functional monomers/functional units will also have a high impact on imprinting factor. However, in the case of 5-FU it is hard to introduce any designed functional units due to 5-FU small size and lack of well-defined functional groups to interact with. The polymer itself (during cross-linking) generates amide groups that form interactions with 5-FU. As a result, we focused on high cross-linking degree. Nevertheless, if the proposed system would be used as DDS for other drugs, the change in cross-linking degree and introduction of additional functional groups will probably be beneficial for the imprinting factor.

  • The drug adsorption was carefully studied, but the data on drug loading and encapsulation efficacy are definitely missing, as they are very important for pharmaceutical application of the system. The adsorption studies description could be minimized and finalized with the conclusion, which describe the system as pharmacologically perspective.

The data about drug loading experiments has now been added. The following information is included in the manuscript: “Unloaded PiPOx-MIP and PiPOx-NIP were immersed for 24 h in 1 mg mL-1 5-FU solution to ensure maximum drug adsorption in the materials structure. the obtained drug-loaded materials were used in subsequent release studies. The amount of drug bonded was calculated based on initial and equilibrium 5-FU concentration in the applied solution”.

This is the first report regarding use of this system for imprinting, therefore in our opinion a detailed study on the adsorption is necessary in this case.

  • Page 9, line 285 “The addition of TCEP only significantly influences the release…” The meaning of only is unclear.

It was supposed to mean that MIPs are influenced and NIPs not. We removed this word, as the sentence without it is still understandable.

  • Page 9, line 302. There are many questions on the release data. (1) The shape of material, its form and specific surface area are unclear and don’t allow to understand the release peculiarities. As well as no data on how much of the drug was loaded. (2) If the drug has enough space to diffuse in the matrix? It seems that not, because you can reach only half of the drug released. (3) What happens with the cross-linked polymer in reductive medium? Swelling, weight loss etc effects are not described. If the reductive medium is relevant to the biological reductive strength? It is also known, that reductive medium is intracellular one. In which form the formulation can be internalized by cells? (4) The amount of the drug released is different, but the triggered release means that the rate of release is different after subjection to the trigger. If the rate is really so different? Please discuss this point more in more detailed manner. (5) It is a little bit strange that release in very acidic medium is the same as at pH 7.4. The protons are reductive themselves as well as 5-FU amide groups should be protonated and favor fast dissolution of the drug. What was the barrier for fast release in this case?

The information about the shape and size of the polymer particles has been included in the manuscript (in different sections). The experimental data about drug loading is now given.

The drug, in our opinion, has enough space to diffuse. However, the limited release may be caused by the difference in conditions between rebinding and release. Rebinding was done in DMF to ensure high loading ratio, however the release was done in aqueous solutions, which might be a cause of limited release.After immersing in release medium the polymer becomes opaque, indicating partial hydrophobic collapse of the drug loaded MIP, which may be responsible for the incomplete release. No swelling or weigh loss was observed for the polymer.

The initial “burst” release is similar and dominating for all materials, with and without TCEP, indicating that it is diffusion controlled release of readily accessible 5-FU. The TCEP influences the maximum release after this initial burst release process finishes as the slower (compared to initial burst release) cleavage of disulfide bond allows further release of less accessible 5-FU.

We do not agree that protons are reductive in this system, they themselves cannot reduce disulfides. The reason of similar release in different pH is probably caused by using TCEP for cleavage. Following the data from Thermo Scientific: “TCEP effectively reduces disulfide bonds over a broad pH range.5 In one experiment, TCEP completely reduced 2,2´dithiodipyridine (2,2´-DTDP) within 30 seconds at 1.5 < pH < 9.0. Above pH 9.0, only 50% reduction occurred. TCEP is a more effective than DTT at pH < 8.0; TCEP will even reduce oxidized DTT.5” (link: https://assets.thermofisher.com/TFS-Assets/LSG/manuals/MAN0011306_TCEP_HCl_UG.pdf). This clearly means that TCEP has similar activity in broad pH ranges which explains the results.

  • It is also important to give some data on the polymer toxicity. At least some from previous studies.

An additional information discussing the biocompatibility of soluble as well as cross-linked materials based on PiPOx has been added in the introduction section as suggested by the Reviewer.

Round 2

Reviewer 1 Report

The authors improved the manuscript as suggested. 

Author Response

We thank the reviewer for the support of our work.

Reviewer 3 Report

  1. My allegations concerned mainly the lack of references and citations of previous works of a similar type. I have never claimed that the work presented is a work replicated previous achievements. In the revised version, the authors included my suggestions regarding the extension of the introductory part to similar previous studies, also the issue of the use of disulfide bond into the polymeric network.

  1. Clarifications regarding the UV VIS measurement methodology have been introduced as suggested. However, I think that the calibration curves used should be published as additional data at Supplementary Material part. Data and observations on fluoracil release kinetics have been significantly expanded, which has increased the readability and level of the work.

  1. The authors extended part 3.4 of the manuscript with suggested appropriate explanations regarding the observed drug release results and reference to earlier results. However, there is lack of specific summary of the possibilities of practical application and an example of the potential usefulness of the presented system in application to selected sample anti-cancer therapy. It seems that the sensitivity to changes in the pH of this system is too low.

I believe that after introducing these small explanations into the text, as well as adding additional material as a Supplementary Part in which you can place UV VIS calibration curves, but also NMR spectra or other measurement data, the work will be suitable for publication.

Author Response

We thank the reviewer for the support of our work.

In the first revised version we had added an appropriate text in the conclusions section to discuss the potential administration of our particles, instead of expanding discussion in part 3.4. Nonetheless, in this second revised version we have also added a brief comment in section 3.4 as follows: ‘These results demonstrate that our system does not function as pH controlled release system while it has potential as reduction-triggered release system.’

The requested data have been provided in the supporting information.

Reviewer 5 Report

The authors have made a great work to revise their MS. It is now acceptable for publication.

Author Response

(The authors gave the same response as above.)
